# Stable Integration of Inducible SPLICS Reporters Enables Spatio-Temporal Analysis of Multiple Organelle Contact Sites upon Modulation of Cholesterol Traffic

**DOI:** 10.3390/cells11101643

**Published:** 2022-05-14

**Authors:** Flavia Giamogante, Lucia Barazzuol, Elena Poggio, Marta Tromboni, Marisa Brini, Tito Calì

**Affiliations:** 1Department of Biomedical Sciences, University of Padova, 35131 Padova, Italy; flaviagiamogante@gmail.com (F.G.); luciabarazzuol@gmail.com (L.B.); marta.tromboni96@gmail.com (M.T.); 2Department of Biology, University of Padova, 35131 Padova, Italy; elena.poggio@unipd.it; 3Padova Neuroscience Center (PNC), University of Padova, 35131 Padova, Italy

**Keywords:** SPLICS, organelle contact sites, split-GFP, stable cell lines, piggyBac system

## Abstract

The study of organelle contact sites has received a great impulse due to increased interest in the understanding of their involvement in many disease conditions. Split-GFP-based contact sites (SPLICS) reporters emerged as essential tools to easily detect changes in a wide range of organelle contact sites in cultured cells and in vivo, e.g., in zebrafish larvae. We report here on the generation of a new vector library of SPLICS cloned into a piggyBac system for stable and inducible expression of the reporters in a cell line of interest to overcome any potential weakness due to variable protein expression in transient transfection studies. Stable HeLa cell lines expressing SPLICS between the endoplasmic reticulum (ER) and mitochondria (MT), the ER and plasma membrane (PM), peroxisomes (PO) and ER, and PO and MT, were generated and tested for their ability to express the reporters upon treatment with doxycycline. Moreover, to take advantage of these cellular models, we decided to follow the behavior of different membrane contact sites upon modulating cholesterol traffic. Interestingly, we found that the acute pharmacological inhibition of the intracellular cholesterol transporter 1 (NPC1) differently affects membrane contact sites, highlighting the importance of different interfaces for cholesterol sensing and distribution within the cell.

## 1. Introduction

Constant rewiring of cellular functions, including transport and storage of nutrients, metabolites, ions, and lipids, as well as protein function and localization, is required to guarantee cell fitness and to cope with the constant energy demand necessary for biosynthesis of macromolecules, cell division, growth, or migration. Signals conveyed at precise locations, i.e., where the membranes of diverse organelles are kept in close apposition by defined factors or protein complexes, are essential to integrate compartmentalized cellular pathways [1,2]. These specialized membrane domains, known as membrane contact sites (MCSs), are considered an important hub where the exchange of nutrients, metabolites, ions, and lipids occurs to deliver either messages of life or death [3]. Their control is mandatory to dictate cellular metabolism flexibility and the study of their reorganization in response to a specific input is therefore important. To have the possibility to investigate these aspects, we generated the first genetically encoded reporter able to detect changes in the ER-mitochondria interface in vitro and in vivo, and named it split-GFP-based contact site sensor (SPLICS) [4]. Later, the SPLICS family was expanded to explore additional relevant contact sites [5] and further optimization allowed for the simultaneous detection of contact sites occurring at different membrane interfaces [5]. SPLICS reporters received a great interest from the scientific community [4,5,6,7,8,9,10,11,12,13,14,15,16,17,18,19,20,21]. Despite their undoubtable utility, possible limitations could arise from the irreversible nature of the split-GFP complementation process and eventually to the possibility that the reconstituted GFP molecule could contribute energy to the tethering of the membranes [22]. However, evidence is convincingly provided that the GFP_1–10_ and β_11_ fragments may interact reversibly when expressed in living cells on two opposing membranes [23]. Having in mind these possible limitations, we report here on the generation of a vector library based on the piggyBac system for stable and inducible expression of SPLICS sensors in a cell line of interest [24]. We tested this system in HeLa cells by introducing SPLICS reporters to detect the endoplasmic reticulum (ER)-mitochondria (MT) interaction (SPLICS_S/L_^ER-MT^), the endoplasmic reticulum (ER)-plasma membrane (PM) interaction (SPLICS_S/L_^ER-PM^), the peroxisomes (PO)-endoplasmic reticulum (ER) interaction (SPLICS_S/L_^PO-ER^), and the peroxisomes (PO)-mitochondria (MT) interaction (SPLICS_L_^PO-MT^). Furthermore, we decided to challenge our HeLa clones expressing inducible SPLICS by blocking cholesterol transport and following membrane contact sites’ reorganization.

Cholesterol is a crucial component of eukaryotic membranes [25]. It is distributed among intracellular membranes, with the majority (60–90%) residing in the plasma membrane and only 0.5–1% in the endoplasmic reticulum. In mammalian cells, the newly synthesized cholesterol travels from the plasma membrane to ER [26,27], while the uptake of the exogenous cholesterol occurs from circulating low-density lipoproteins (LDLs)-carried cholesterol [28]. Cholesterol is liberated from cholesteryl esters by acid lipase in the endocytic compartment, to be delivered to late endosome/lysosomes [18]. To date, many efforts have been devoted to deep investigation of intracellular cholesterol dynamics and, recently, the crucial role of lysosomes in cholesterol trafficking [29] emerged. Lysosomal cholesterol moves to the ER and from here to peroxisomes via the ER-resident extended synaptotagmin-1, 2 and 3 and the peroxisomal PI(4,5)P2 [30,31]. Interestingly, it has been reported that upon inhibition of the Niemann-Pick C protein 1 (NPC1), ER-lysosome cholesterol trafficking is inhibited, while the lysosomes-mitochondria membrane contact sites were expanded by the activation of the sterol-binding protein STARD3 [32]. 

In this respect, we decided to exploit the potentiality of these cell lines to follow the rearrangement of organelle contact sites upon lysosomal cholesterol accumulation by treating cells with U18666A, which blocks NPC1 by binding to the sterol-sensing domain of NPC1. Interestingly, under these conditions, we detected a statistically significant increase in the contact sites between the ER-PM that was paralleled by a strong decrease in the number of ER-PO contact sites, suggesting a direct relationship between the regulation of intracellular cholesterol fluxes and ER-PM and ER-PO contact sites remodeling. Differently, the ER-MT as well as the PO-MT contact sites remained unaffected by the treatment, suggesting that under this condition, a specific contact site-dependent route was engaged. 

## 2. Materials and Methods

### 2.1. Cloning Plasmid Construction

In order to generate the Expression Vector Library, the candidate cDNAs (SPLICS_S/L_^ER-PM^, SPLICS_S/L_^PO-ER^, SPLICS_S/L_^ER-MT^, SPLICS_L_^PO-MT^) were amplified from pcDNA3 containing sequences and cloned into a pENTR2B donor vector by Gateway^®^ BP Clonase™ II Enzyme (Thermo Fisher Scientific, Waltham, MA, USA) following the manufacturer’s instructions. Then, the transgenes were Gateway Technology-cloned into the destination vector containing PB-TRE-DEST by Gateway^®^ LR Clonase™ II Enzyme (Thermo Fisher Scientific, Waltham, MA, USA) following the manufacturer’s instructions. 

### 2.2. Cell Line

HeLa cells (ATCC) were grown in Dulbecco’s Modified Eagle’s Medium (DMEM) (Thermo Fisher Scientific, Waltham, MA, USA; 41966-029) high glucose, 110 mg/L sodium pyruvate supplemented with 10% (*v*/*v*) Fetal Bovine Serum (Thermo Fisher Scientific, Waltham, MA, USA; Cat# 10270-106), 100 units per ml Penicillin and 100 μg/mL Streptomycin (Penicillin–Streptomycin solution 100×) (EuroClone; Cat# ECB3001D). Cells were maintained at 37 °C in a 5% CO_2_ atmosphere.

### 2.3. Stable Transfection

Stable transgenic HeLa expressing SPLICS_S/L_-P2A probes were generated by transfecting cells with two piggyBac (PB) transposon plasmids, pPB-rtTAM2-IresNeo and pPB-TRE-Dest-SPLICS, and PB transposase expression vector p-HyPBase. HeLa cells were transfected using Lipofectamine 2000 Transfection Reagent (Thermo Fisher Scientific, Waltham, MA, USA, Cat. 11668019) in accordance with the manufacturer’s instructions. Briefly, HeLa cells were seeded in a 6-well plate (Euroclone, Milan, Italy) at 80–90% confluence and transfected with Lipofectamine 2000 Transfection Reagent plus 0.7 ug of pPB-rtTAM2-IresNeo, 1.5 ug of pPB-TRE-Dest-SPLICS, and 0.4 ug of p-HyPBase. Six hours after transfection, the medium was replaced with a fresh medium. After 48 h, 2 mg/mL geneticin (Thermo Fisher Scientific, Waltham, MA, USA, Cat. 10131027) was added to the cells to select stable transfected cell clones. Once selected, 0.1 mg/mL of geneticin was added to maintain stable transfected cell clones.

### 2.4. Immunicytochemistry and Filipin Staining

To assess organelles morphology under U18666A treatment, HeLa cells were seeded at 60–80% confluence onto 13 mm diameter glass coverslips. After overnight induction, the medium was replaced with a fresh medium containing 2 µg/mL of U18666A (Merck KGaA, Darmstadt, Germany, Cat. 662015). After 18-h treatment, the cells, plated on 13 mm glass coverslips, were fixed for 20 min in a 3.7% (*v*/*v*) formaldehyde solution (Merck KGaA, Darmstadt, Germany; Cat# F8775). Cells were then washed three times with D-PBS. Cell permeabilization was performed by 10 min incubation in 0.3% Triton X-100 Bio-Chemica (PanReac AppliChem, Milan, Italy; A1388) in D-PBS, followed by washing three times in 1% gelatin/D-PBS (Type B from bovine skin) (Merck KGaA, Darmstadt, Germany; G9382) for 15 min at room temperature (RT). The coverslips were then incubated for 90 min at RT with the specific primary antibody diluted in D-PBS (1:50 Tom20: Santa Cruz Biotechnology, Dallas, TX, USA, Cat#sc-11415, 1:100 KDEL: Abcam, Cat#ab2898, 1:50 PMP70: Merck KGaA, Darmstadt, Germany Cat#SAB4200181). Three washes with 1% gelatine/D-PBS were performed to remove the excess primary antibody. Staining was revealed by the incubation with a dilution 1:100–1:200 in D-PBS of specific Alexa Fluor secondary antibodies (Thermo Fisher Scientific, Waltham, MA, USA: Goat anti-Rabbit IgG Alexa Fluor 594, A32740; Goat anti-Mouse IgG Alexa Fluor 633, A-31553; Goat anti-Mouse IgG Alexa Fluor 488; A32723; Goat anti-Rabbit IgG Alexa Fluor 488, A32731) for 45 min at room temperature. After three additional washes with 1% gelatine/D-PBS, plasma membrane signal was detected by an mCherry-CaaX(Hras) plasmid, a gift from Rob Parton (Addgene plasmid # 108886). Cholesterol accumulation was evaluated by Filipin III staining (Merck KGaA, Darmstadt, Germany; F476) following the manufacturer’s instructions. Briefly, after 18-h U18666A treatment, the cells, plated on 13 mm glass coverslips, were fixed for one half hour in a 4% (*v*/*v*) paraformaldehyde/DPBS solution (Merck KGaA, Darmstadt, Germany; Cat# 158127). The coverslips were then incubated for two hours at RT in the dark, followed by three washes in DPBS. Coverslips were mounted using Mowiol 4–88 (Merck KGaA, Darmstadt, Germany).

### 2.5. Doxycycline Induction 

Twelve hours before induction, stable transgenic HeLa expressing SPLICS_S/L_-P2A probes were seeded at 60–80% confluence onto 13 mm diameter glass coverslips (for immunofluorescence contact sites analysis) or in 6-multiwell plates (for western blotting analysis). The medium was replaced with a fresh medium before adding increasing concentrations of doxycycline (Merck KGaA, Darmstadt, Germany, Cat. D3447), from 1 ng/mL to 100 ng/mL for SPLICS_S/L_-P2A^ER-PM^, SPLICS_S/L_-P2A^ER-MT^_,_ and SPLICS_L_-P2A^PO-MT^; and from 10 ng/mL to 500 ng/mL for SPLICS_S/L_-P2A^PO-ER^. Twenty-four hours post induction, the cells, plated on 13 mm glass coverslips, were fixed for 20 min in 3.7% (*v*/*v*) formaldehyde solution (Sigma-Aldrich; Cat# F8775). Cells were then washed three times with D-PBS (Euroclone, Milan, Italy). Mitochondria morphology of stable transgenic HeLa expressing SPLICS_S/L_^ER-MT^ and SPLICS_L_^PO-MT^ was detected by MitoTracker™ Red CMXRos (Invitrogen, Cat. M7512) in accordance with the manufacturer’s instructions. Briefly, before fixing, cells were washed in Hank’s Balanced Salt Solution (Thermo Fisher Scientific, Waltham, MA, USA) and incubated with 150 nM MitoTracker™ Red (Invitrogen), for 20 min at 37 °C in a 5% CO_2_ atmosphere. Coverslips were mounted using Mowiol 4–88 (Sigma) (Invitrogen).

### 2.6. U18666A Treatment

Twelve hours before induction, stable transgenic HeLa expressing SPLICS_S_-P2A probes were seeded at 60–80% confluence onto 13 mm diameter glass coverslips. Medium was replaced with fresh medium before adding different doxycycline concentration (10 ng/mL for SPLICS_S_^ER-MT^, SPLICS_S_^ER-PM^, and SPLICS_L_^PO-MT^; and 500 ng/mL for SPLICS_S_^PO-ER^). After 24-h induction, the medium was replaced with a fresh medium without doxycycline but containing 2 µg/mL of U18666A (Merck KGaA, Darmstadt, Germany, Cat. 662015). After 18-h treatment, the cells, plated on 13 mm glass coverslips, were fixed for 20 min in a 3.7% (*v*/*v*) formaldehyde solution (Merck KGaA, Darmstadt, Germany; Cat# F8775). Cells were then washed three times with D-PBS. Coverslips were mounted using Mowiol 4–88 (Merck KgaA, Darmstadt, Germany).

### 2.7. Confocal Microscopy and Image Analysis 

Cells were imaged with a Leica TSC SP5 inverted confocal microscope, using either a HCX PL APO 100×/numerical aperture 1.40–0.60 or a HCX PL APO ×100/numerical aperture 1.4 oil-immersion objective. Images were acquired by using the Leica AS software. To count ER-MT, ER-PM, PO-MT, and PO-ER contacts, a complete Z-stack of the cell was acquired every 0.29 μm and processed using ImageJ National Institutes of Health (NIH). Images were first convolved, and the cells were selected using the freehand selection of ImageJ in the drawing/selection polygon tool and then processed using the “Quantification 1” plugin (https://github.com/titocali1/Quantification-Plugins accessed on 10 December 2021). A 3D reconstruction of the resulting image was obtained using the Volume J plugin (https://github.com/titocali1/Quantification-Plugins accessed on 10 December 2021). A selected face of the 3D rendering was then thresholded and used to count short and long contact sites through the “Quantification 2” plugin (https://github.com/titocali1/Quantification-Plugins accessed on 10 December 2021).

### 2.8. Western Blot

Twenty-four hours post induction, cellular extracts of stable transgenic HeLa expressing SPLICS_S/L_-P2A probes were western blotted to evaluate GFP_1–10_ or YFP_1–10_ protein expression. Cells were lysed in RIPA Buffer (50 mM Tris-HCl pH 7.4, NaCl 150 mM, 1% Triton X-100, 0.5% sodium deoxycholate, 10 mM EDTA, 0.1% SDS, 1 mM DTT, 2× Protease Inhibitor Cocktail (Merck KGaA, Darmstadt, Germany, Cat# P8340)) for 20 min. Extracted proteins were quantified by Bradford assay (Bio-Rad, Hercules, California, USA Cat# 500-0205), resolved on SDS-PAGE in 12% SDS/PAGE Tris- HCl polyacrylamide gel, and then transferred to PVDF membranes (Bio-Rad, Hercules, California, USA) using Trans-Blot^®^ Turbo™ Transfer System (Bio-Rad, Hercules, California, USA). Membranes were blocked with 5% *w/v* non-fat dried milk (NFDM) in TBST (20 mM Tris-HCl, pH 7.4, 150 mM NaCl, 0.05% Tween-20) and incubated overnight with the specific primary antibody at 4 °C. Signal was detected by incubation with secondary horseradish peroxidase-conjugated anti-rabbit or anti-mouse IgG antibodies for 1 h at room temperature followed by incubation with the chemiluminescent reagent Luminata Classico HRP substrate (Merck Millipore, Cat# WBLUO500), monoclonal anti-GFP 1:500 (Santa Cruz, Cat. 9996) and monoclonal anti-ßeta Tubulin 1:2000 (Cell Signalling, Cat. 2128).

### 2.9. Statistical Analysis

All statistical analyses were performed using GraphPad Prism version 8.00 for Mac OS X (La Jolla, CA, USA). Results shown are mean values ± SEM. Statistical comparisons were performed using Student’s unpaired two-tailed *t*-test. Statistical significance threshold was set at *p* < 0.05. The exact values of *n* and their means are indicated in the figure legends. * *p* ≤ 0.05, ** *p* ≤ 0.01, *** *p* ≤ 0.001, **** *p* ≤ 0.0001.

## 3. Results

### 3.1. Vector Library Generation and Generation of HeLa Polyclonal Stable Cell Lines

The vector library including all the pPB-TRE-Dest-SPLICS plasmids was generated. The first step involves the insertion of the PCR fragment of the SPLICS reporter of interest flanked by two attB sequences into a pPB-TRE-Des vector containing two homologous attP sites by using the gateway technology catalyzed by the BP clonase enzyme. The SPLICS reporters to be used as a template are available on Addgene (https://www.addgene.org/Tito_Cali/ accessed on 2 February 2021). The recombination between the attB sites on the PCR product and the attP sites on the donor vector will generate attL sites and release the suicide cassette contained in the donor vector (Figure 1A). 

A second recombination catalyzed by the LR clonase between the attL sites in the donor vector and the attR sites contained in the destination vector (pPB-TRE-Dest) generated the expression clone (pPB-TRE-Dest-SPLICS) containing the SPLICS reporter flanked with attB sites and released the suicide cassette contained into the destination vector (Figure 1A). A library of SPLICS expression vectors has been generated for the reporters involving the ER-PM, the ER-PO, the ER-mitochondria, and the PO-mitochondria contact sites. The SPLICS reporters are composed of two fragments of the split-GFP variant, GFP1-10 (or its spectral variant YFP1-10), and β11, targeted to the outer face of the organelles through the use of minimal targeting sequences [4,5,33]. They are available in two variants known as SPLICS Short and SPLICS Long that can detect contact sites and/or organelle proximities in the 8–10 nm or in the 40–50 nm range, respectively. This sensitivity is achieved by the addition of a flexible Gly-Ser linker of different lengths placed between the targeting sequence and the β11 fragment. The different spectral variants of the GFP protein can be used to generate GFP or YFP-based single reporters or exploited to generate a third generation of SPLICS reporters to simultaneously visualize two different contact sites within the same cell upon reconstitution with the same invariant β11 fragment [5]. These vectors have been used to co-transfect HeLa cells for stable integration in combination with two additional plasmids encoding the Tet-On repressor and antibiotic-resistance (pPB-rtTAM2-IresNeo) and the piggyBac transposase (referred to as the helper PBase plasmid p-HyPBase) (Figure 1B). Stable integration is ensured by the insertion into one of the TTAA sequences distributed throughout the genome thanks to the recombination between two terminal repeats (PB sequences) present in the pPB-rtTAM2-IresNeo and in the pPB-TRE-Dest-SPLICS, recognized by the transiently expressed PB transposase to allow a quick transfer of the gene of interest. In the present work, the library consisting of the SPLICS_S/L_^ER-PM^, SPLICS_S/L_^PO-ER^, SPLICS_S/L_^ER-MT^, and SPLICS_L_^PO-MT^ reporters was generated and will be expanded to cover all the available contact sites (Figure 1C). Since we have previously observed that the PO-MT SPLICS showed no significant differences in terms of number and distribution of short- and long-range interactions [5], here we decided to develop the piggyBac system only for the long reporter, while both the short and the long SPLICS were taken into account for the PO-ER in order to better investigate the difference in the pattern of contacts that we had previously observed [5].

### 3.2. Western Blotting of Protein Expression in HeLa SPLICS_S/L_^ER-MT^ and SPLICS_L_^PO-MT^, SPLICS_S/L_^ER-PM^ and SPLICS_S/L_^PO-ER^ Stable Cell Lines

As previously shown in Vallese et al. [5], SPLICS probes are correctly expressed on targeted organelle surfaces, do not alter organelles’ morphology and shape, and permit the detection of contact sites occurring at short (in the 8–10 nm range) and long (in the 40–50 nm range) distance. The development of stably transfected HeLa cells expressing a palette of SPLICS sensors represent a good tool to take advantage of this technology and the application of an inducible system to titer the SPLICS sensors’ expression offers several advantages. An optimization protocol has been performed to check the SPLICS expression, exploiting the Tet-On system present in the pPB-TRE-Dest-SPLICS plasmid. 

We firstly created HeLa cell lines expressing the SPLICS and, after selection of polyclonal stable transfected cells, we tested the efficiency of the Tet-On system using increasing concentrations of doxycycline (1, 10, 50, 100, and 500 ng/mL, as indicated in the different panels of Figure 2) for 24-h treatment in HeLa polyclonal stable cell lines expressing SPLICS_S/L_^ER-PM^, SPLICS_S/L_^PO-ER,^, SPLICS_S/L_^ER-MT^, and SPLICS_L_^PO-MT^.

The doxycycline concentration required to obtain ideal expression of the reporters was titrated by western blotting analysis and immunoblotting detection has been performed only against the GFP/YFP_1–10_ portion in all SPLICS probes due to the availability of an antibody for its detection. Equimolar production of the two split-GFP fragments is ensured by the P2A sequence [33]. In all the considered reporters, the GFP/YFP_1–10_ portion is not expressed in the absence of doxycycline treatment and under low doxycycline concentration (in line with the immunofluorescence data reported below), demonstrating the efficiency of the Tet-On system (Figure 2). Starting from 10 ng/mL of doxycycline, the signal of the GFP/YFP_1–10_ portion becomes detectable in all the considered probes (albeit in the case of SPLICS_S/L_^PO-ER^ probe, the signal is barely detectable) and its expression increases with higher doxycycline concentrations, suggesting that the reporters are expressed in a dose-dependent manner, as also shown by the quantification (Figure 2). Considering that the short and long contacts in all SPLICS reporters are due to different lengths of the linker fused with ß_11_ portion, the western blot analysis of the GFP/YFP_1–10_ portion revealed no differences in the molecular weight of the short and long probes, as expected. A difference in the molecular weight is instead appreciated between YFP and GFP_1–10_ portions according to the length of their targeting sequence to the specific organelle. In particular, the YFP_1–10_ portion is targeted to the plasma membrane for SPLICS_S/L_^ER-PM^ and to the ER membrane for SPLICS_S/L_^PO-ER^, respectively. Instead, the GFP_1–10_ portion is targeted to the outer mitochondrial membrane for both SPLICS_S/L_^ER-MT^ and SPLICS_L_^PO-MT^ (uncropped gels are shown in Appendix A). 

These results indicate that SPLICS expression can be finely tuned dose-dependently and that the optimal concentration required to detect specific contact sites can be adjusted according to the specific reporter, indeed, for instance, high expression of the SPLICS_L_^PO-MT^ probe occurs at a low doxycycline concentration (50 ng/mL) while that of the SPLICS_S/L_^PO-ER^ requires a higher amount (100–500 ng/mL). 

### 3.3. Short and Long ER-Plasma Membrane and ER-Peroxisomes Interactions in HeLa Polyclonal Stable Cell Lines Express Inducible SPLICS Sensors

Once validated in terms of SPLICS expression levels, the efficiency of the Tet-On system was further tested by monitoring GFP/YFP fluorescence reconstitution in confocal images acquired from the HeLa polyclonal stable cell lines expressing SPLICS_S/L_^ER-PM^, SPLICS_S/L_^PO-ER^, SPLICS_S/L_^ER-MT^, and SPLICS_L_^PO-MT^. The range of increasing concentrations of doxycycline used in the western blotting analysis was applied for 24-h treatment. Figure 3 shows HeLa polyclonal stable cell lines expressing SPLICS_S/L_^ER-PM^ and SPLICS_S/L_^PO-ER^ upon induction with doxycycline 1, 10, and 100 ng/mL and 10, 100, and 500 ng/mL, respectively. As expected, both SPLICS_S/L_^ER-PM^ and SPLICS_S/L_^PO-ER^ are expressed in a dose-dependent manner (Figure 3A,C). 

Notably, no signal is detected in untreated cells and at low doxycycline concentrations, specifically 1 ng/mL for SPLICS_S/L_^ER-PM^ and 10 ng/mL for SPLICS_S/L_^PO-ER^, demonstrating the efficiency of the Tet-On system. At 100 ng/mL of doxycycline, in some cells, the expression of SPLICS_S_^ER-PM^ Short induces the alteration of the typical punctate pattern [5,34] of ER-PM interactions, as indicated by the white arrow in Figure 3A. However, the same effect could not be observed for SPLICS_L_^ER-PM^ Long and for SPLICS_S/L_^PO-ER^, highlighting the importance of finely tuning the expression of each SPLICS sensor and the fact that in general, lower concentrations should be preferred.To assess whether the number of SPLICS puncta could be affected by the doxycycline concentration, we decided to quantify the number of contacts per cell for SPLICS_S/L_^ER-PM^ at 10 ng/mL and 100 ng/mL doxycycline (Figure 3B), and 100 ng/mL and 500 ng/mL doxycycline for SPLICS_S/L_^PO-ER^ (Figure 3D). Interestingly, when we compared the mean dots at 10 and 100 ng/mL of doxycycline, no significant differences were detected in SPLICS_S/L_^ER-PM^ (mean ± SEM: 216.5 ± 9.25 *n* = 28 and 236.3 ± 10.68 *n* = 25 at 10 and 100 ng/mL of doxycycline, respectively, for SPLICS_S_^ER-PM^; 266.9 ± 21.42 *n* = 23 and 243.1 ± 16.05 *n* = 25 at 10 and 100 ng/mL of doxycycline, respectively, for SPLICS_L_^ER-PM^). The same was true for the SPLICS_S/L_^PO-ER^: no significant difference in the number of puncta was detected at different doses of doxycycline (mean ± SEM: 93.91 ± 7.48 *n* = 27 and 109.0 ± 10.24 *n* = 26 at 100 and 500 ng/mL of doxycycline for SPLICS_S_
^PO-ER^, respectively; 111.7 ± 7.11 *n* = 29 and 123.8 ± 7.32 *n* = 26 at 100 and 500 ng/mL of doxycycline for SPLICS_L_-P2A^PO-ER^, respectively). 

These data demonstrate that the mean contacts per cell is not affected by doxycycline concentration, and that fine titration should be applied to each SPLICS to achieve homogeneous expression. Interestingly, the long ER-PM and PO-ER contacts are increased compared to the short ones (Figure 3E,F), confirming that HeLa polyclonal stable cell lines expressing SPLICS_S/L_^ER-PM^ and SPLICS_S/L_^PO-ER^ upon induction with doxycycline are a suitable model to explore the physiological meaning of this difference.

### 3.4. Short and Long Mitochondria-ER and Mitochondria-Peroxisomes Interactions in HeLa Polyclonal Stable Cell Lines Express Inducible SPLICS Sensors

Analogously to that shown above for the ER-PM and the PO-ER contacts, we have generated HeLa polyclonal stable cell lines expressing SPLICS_S/L_^ER-MT^ and SPLICS_L_^PO-MT^ to detect ER-mitochondria and peroxisome-mitochondria tethering. As shown in Figure 4, after 24 h of doxycycline treatment, both SPLICS_S/L_ ^ER-MT^ and SPLICS_L_^PO-MT^ are expressed dose-dependently. 

We observed that in some cells, upon incubation with 100 ng/mL of doxycycline, the expression of the SPLICS_S_^ER-MT^ was very high and the typical dot pattern of the SPLICS signal was altered, as indicated by the white arrowheads on the images (Figure 4A). Similarly, at 50 ng/mL of doxycycline, the correct SPLICS_L_-P2A^PO-MT^ puncta signal is substantially altered (white arrowheads, Figure 4C).

Similarly, to the above SPLICS_S/L_^ER-PM^ and SPLICS_S/L_^PO-ER^, we decided to quantify the number of contacts per cell to verify whether doxycycline concentration could affect the number of dots. The quantification was performed at 10 and 100 ng/mL of doxycycline for SPLICS_S/L_^ER-MT^ and no significant differences were reported between the two doxycycline concentrations, demonstrating that doxycycline does not alter the number of dots (Figure 4B) (value, means ± SEM: SPLICS_S_-P2A^ER-MT^ 97.48 ± 8.62 *n* = 27 and 89.88 ± 5.13 *n* = 25 at 10 and 100 ng/mL of doxycycline, respectively; SPLICS_L_-P2A^ER-MT^ 160.7 ± 17.40 *n* = 23 and 138.3 ± 7.91 *n* = 26 at 10 and 100 ng/mL of doxycycline, respectively). Even in this case, when we compared the mean number of short and long ER-mitochondria contacts, significant differences were observed between them (Figure 4E,F; also comparing the number of different membrane contact sites assessed), as already reported [4,5]. 

In the case of SPLICS_L_-P2A^PO-MT^, contact quantification was only performed at 10 ng/mL of doxycycline, since at 50 ng/mL of doxycycline, the correct SPLICS puncta signal is altered (Figure 4B). The mean peroxisomes-mitochondria contacts number is in line with the values previously obtained with the second generation of SPLICS probe [5], as also reported in the table in Figure 4F (value, means ± SEM: SPLICS_L_-P2A^PO-MT^ 50.2 ± 2.48 *n* = 48). 

To better clarify whether the alteration of the puncta signal of reconstituted SPLICS_S/L_^ER-MT^ and SPLICS_L_-P2A^PO-MT^ observed in some cells at the higher doxycycline concentration could be due to altered mitochondrial morphology because of excessive expression of the mitochondria targeted OMM-GFP_1–10_ portion, HeLa polyclonal stable cell lines expressing these mitochondria contact sites reporters were loaded with Mitotracker Red. 

We observed that the alteration of the typical dot pattern of the SPLICS signal was often reflected by gross changes in the mitochondrial network (white arrows in Figure 4A,C). Cells with a lower level of induction show no alterations in the mitochondrial morphology, as indicated by the asterisks. Notably, the alteration of the SPLICS signal was not observed for the long-range ER-MT contact sites detected with the SPLICS_L_^ER-MT^ (Figure 4A). As for the SPLICS_L_^PO-MT^ expression, the mitochondrial morphology was deeply altered upon incubation with 50 ng/mL of doxycycline, mitochondria being fragmented and collapsed in the perinuclear region in most of the cells as indicated by arrows (Figure 4C). On the other hand, at 10 ng/mL of doxycycline, the puncta pattern of the mitochondria-peroxisomes interaction is preserved without visible changes in the filamentous ultrastructure of mitochondria. These observations, together with the shorter incubation times tested in Appendix A, indicate that the amount of expressed SPLICS should be accurately evaluated under each specific condition and that this system is tunable to choose the proper dosage of SPLICS expression in order to get a good fluorescent signal without impacting on organelles’ architecture.

To better understand the dynamics of SPLICS induction, we have quantified the increase in the percentage of SPLICS-expressing cells in response to different doxycycline concentrations and the SPLICS fluorescence intensity per cell to check whether the SPLICS signal per se remains unchanged upon increasing doxycycline concentrations. As shown in Appendix A, no differences were found in the fluorescence intensity of the SPLICS signal while the percentage of SPLICS-expressing cells increased with increasing concentrations of doxycycline. 

### 3.5. Effect of U18666A Treatment of Stable HeLa Cell Lines Expressing SPLICS_S_^ER-PM^ and SPLICS_S_^PO-ER^, SPLICS_S_^ER-MT^, and SPLICS_L_^PO-MT^


Once the best doxycycline concentration to induce the SPLICS probes expression for each HeLa polyclonal stable cell line was identified, we decided to investigate the possible reorganization of organelles’ contact sites upon blocking the NPC1 cholesterol transporter. Except for the PO-MT SPLICS (referred to as SPLICS_L_-P2A^PO-MT^), only short-range SPLICS reporters were chosen for this experiment since most of the cholesterol trafficking occurs within the 30 nm range [35,36]. In particular, HeLa cells expressing SPLICS_S_^ER-PM^, SPLICS_S_^ER-MT^, and SPLICS_L_^PO-MT^ were treated with 10 ng/mL of doxycycline, while 500 ng/mL doxycycline was used for the induction of SPLICS_S_^PO-ER^. Twenty-four hours post-induction, 2 µg/mL of the inhibitor of the NPC1 cholesterol transporter U18666A were added to the cells and incubated for 18 h. Interestingly, U18666A significantly affected the ER-PM and PO-ER contacts in the opposite direction, while it had no effect on the ER-MT and PO-MT interactions (Figure 5). 

In particular, U18666A treatment induced a statistically significant increase in the number of ER-PM interactions, as quantified in Figure 5A (value, means ± SEM: SPLICS_S_^ER-PM^ 223.0 ± 10.67 *n* = 23 and 261.4 ± 11 *n* = 29 in untreated and U18666A-treated cells, respectively). Furthermore, U18666A treatment induced a redistribution of the SPLICS_S_^ER-PM^ puncta towards the perinuclear region compared to untreated cells (Figure 5A), a finding that certainly requires further investigation. Conversely, PO-ER interactions were dramatically decreased upon U18666A treatment compared to the control (value, means ± SEM: SPLICS_S_^PO-ER^ 99.90 ± 9.08 *n* = 24 and 45.41 ± 6.49 *n* = 28 in untreated and U18666A-treated cells, respectively, Figure 5B). Finally, no changes were observed in HeLa cell lines expressing SPLICS_S_^ER-MT^ (Figure 5C) and SPLICS_L_^PO-MT^ (Figure 5D) following U18666A treatment (value, means ± SEM: SPLICS_S_^ER-MT^ 81.68 ± 5.54 *n* = 31 and 94.91 ± 5.86 *n* = 39 in untreated and U18666A-treated cells, respectively; SPLICS_L_^PO-MT^ 50.67 ± 3.16 *n* = 24 and 48.62 ± 2.54 *n* = 37 in untreated and U18666A-treated cells, respectively), suggesting that the cells responded to the block of cholesterol trafficking by reorganizing the ER-PM and PO-ER contacts, leaving the ER-MT and PO-MT interactions unaffected (see also Figure 5H). To eliminate the possibility that the changes observed in the SPLICS signal upon incubation with U18666A were due to intracellular rearrangement of the organelles or changes in their morphology, co-staining experiments for endogenous organelle markers were performed in HeLa cells in the absence (Figure 5E) or in the presence (Figure 5F) of U18666A treatment. No gross alteration of the organelles’ morphology was detected, as revealed by the immunocytochemistry analysis performed using KDEL, TOM20, and PMP70 primary antibodies to stain the ER, mitochondria, and peroxisomes, respectively. The PM signal was revealed by the expression of the fluorescent mCherry-CAAX targeted to the PM [37]. Figure 5G shows cholesterol accumulation in HeLa cells after cholesterol staining with Filipin III.

To eliminate the possibility that the increase in ER-PM and the decrease in PO-ER contacts numbers observed in Figure 5A upon U18666A treatment were due to changes in the organelle morphology induced by SPLICS_S_-P2A^ER-PM^ and SPLICS_S_-P2A^PO-ER^ reporters, immunofluorescence analysis was performed in either untreated or U18666A-treated HeLa polyclonal stable cell lines expressing them. As shown in Appendix A, the morphology of ER, peroxisomes, and PM, was not drastically altered by U18666A treatment or by the expression of SPLICS (compare untransfected cells in the same field for different conditions). Appendix A also shows that the morphology of the organelles in cells expressing the short and the long variant of the SPLICS-P2A^PO-ER^ reporter is quite similar, indicating that the different pattern of contacts observed with the short and the long SPLICS-P2A^PO-ER^ (Figure 3C) is specifically dependent on the degree of proximity, and not to different ER or peroxisomes’ morphology or distribution. 

## 4. Discussion

Contact sites between membranous organelles have been a matter of intense research for decades. Although their structures, biological functions, and behavior has been extensively characterized, the lack of efficient and reliable reporters to quantify and visualize organelle contact sites in vitro and in vivo hindered the possibility to gain important insights into their function, as well as to explore their alteration in disease conditions. We designed for the first time split-GFP-based contact sites sensors for the detection of ER-mitochondria contact sites, and tested them in vitro and in vivo [4]. An expanded palette of improved reporters with equimolar production of the two split-GFP fragments has also been generated and proved efficient in quantifying ER-PM, ER-PO, and PO-MT contact sites, either as single reporters or as dual reporters, i.e., with the possibility to quantify different contact sites within the same cell [5,33].

Here, we report on the generation of a vector library for stable integration of inducible SPLICS reporters for spatiotemporal analysis of multiple organelles’ contact site behavior upon cell manipulation. A set of vectors, namely, the SPLICS_S/L_^ER-MT^, SPLICS_S/L_^ER-PM^, SPLICS_S/L_^PO-ER^, and SPLICS_L_^PO-MT^ were used to generate stable HeLa cell lines, and their efficiency and response to cell treatment was tested as a proof of principle experiment. Our results showed that a precise threshold for the induction of SPLICS is present, beyond which the signal starts to become inappropriate, and the morphology of mitochondria starts to be affected (see the comments on Figure 5 in the Results section). An interesting finding comes from the multiple analyses of the different contact sites considered upon treatment with a cholesterol trafficking impairing compound. With the aim of causing a general rewiring of different contact sites, we chose the NPC1 inhibitor U18666A [32] for its ability to affect cholesterol homeostasis by blocking its transfer from lysosomes to the ER. Indeed, while the pool of HDL-cholesterol is internalized at the plasma membrane (PM) and actively transported to the ER through contact sites by proteins of the Aster family (GRAMD1B) [38], the LDL-derived cholesterol is acquired via receptor-mediated endocytosis [39] and transported to the lysosome [40]. The distribution of cholesterol from lysosomal membranes to downstream subcellular locations occurs through heterotypic contacts with organelles such as the PM, lysosomes, ER, and mitochondria (21, 25). From the lysosomal carriers NPC1 and LIMP-2, cholesterol is transported to the ER via direct contacts [31,32]. Sterol exchange at the membrane contact sites between lysosomes and other organelles [41,42] is therefore fundamental for the correct distribution of cholesterol within all the cellular compartments. We have monitored the rewiring of contact sites between the ER-PM, ER-PO, ER-MT, and PO-MT and stumbled upon the interesting finding that U18666A treatment selectively remodeled the ER-PM interface, probably in response to the decreased lysosome-ER cholesterol transfer and, concomitantly, the contact sites of the ER with peroxisomes dramatically dropped. Of note, the ER-mitochondria and the peroxisome-mitochondria interface remained unaltered, highlighting the strong specificity and contact site-dependent response to a given treatment. 

## 5. Conclusions

We believe that the newly developed vector library for the generation of stable cell lines expressing SPLICS reporters will be of pivotal importance to unwinding the behavior of multiple organelles’ contact sites in time and space under a series of physiopathological stimuli. An inducible doxycycline-controlled SPLICS expression not only offers the considerable advantage of fine tuning their expression, thus mitigating any potential artifact due to an excessive level of expression that could induce alteration in organelles’ morphology, in particular mitochondria, but also avoids any possible pitfalls due to artificial organelles’ tethering, potentially caused by the split-GFP complementation process.

## Figures and Tables

**Figure 1 cells-11-01643-f001:**
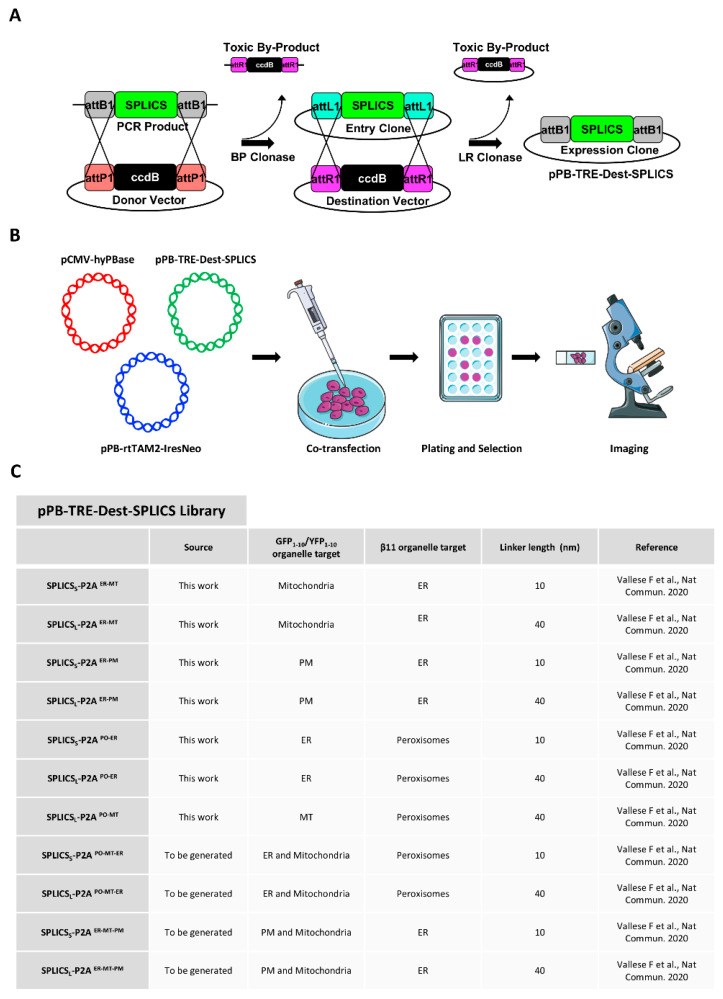
Graphical representation of fundamental steps to generate stable cell lines expressing SPLICS_S/L_-P2A probes. (**A**) Cartoon representation of Gateway system (Invitrogen) to obtain the pPB-TRE-Dest-SPLICS library. (**B**) Workflow to stably transfect HeLa cells with SPLICS_S/L_-P2A probes. Three plasmids were used to stably transfect HeLa cells: p-HyPBase encoding the piggyBac (PB) transposase, pPB-rtTAM2-IresNeo containing the PB sequences and encoding the TET-ON repressor and antibiotic-resistance sequence; and the pPB-TRE-Dest-SPLICS containing the PB sequences and encoding the inducible TET-ON system to produce different SPLICS reporters. (**C**) Table summarizing SPLICS information on organelle target GFP portions and linkers [5].

**Figure 2 cells-11-01643-f002:**
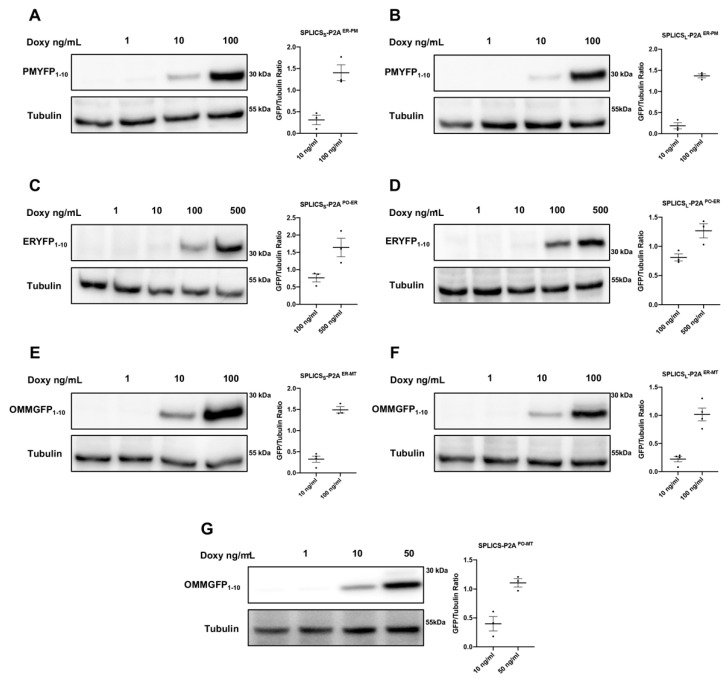
Western blotting analysis of HeLa polyclonal stable cell lines expressing SPLICS_S/L_^ER-PM^ and SPLICS_S/L_^PO-ER^, SPLICS_S/L_^ER-MT^ and SPLICS_L_^PO-MT^. (**A**–**D**) Expression levels YFP1-10 proteins in HeLa stably transfected with SPLICS_S/L_^ER-PM^ or SPLICS_S/L_^PO-ER^ and incubated with increasing concentrations of doxycycline for 24 h. (**E**–**G**) Expression levels of OMMGFP1-10 proteins in HeLa expressing SPLICS_S/L_^ER-MT^ or SPLICS_L_^PO-MT^ upon incubation with increasing concentrations of doxycycline for 24 h. An anti-GFP antibody was used to detect both YFP and GFP. Equal amount of total loaded lysate was verified by incubation with anti-β Tubulin antibody.

**Figure 3 cells-11-01643-f003:**
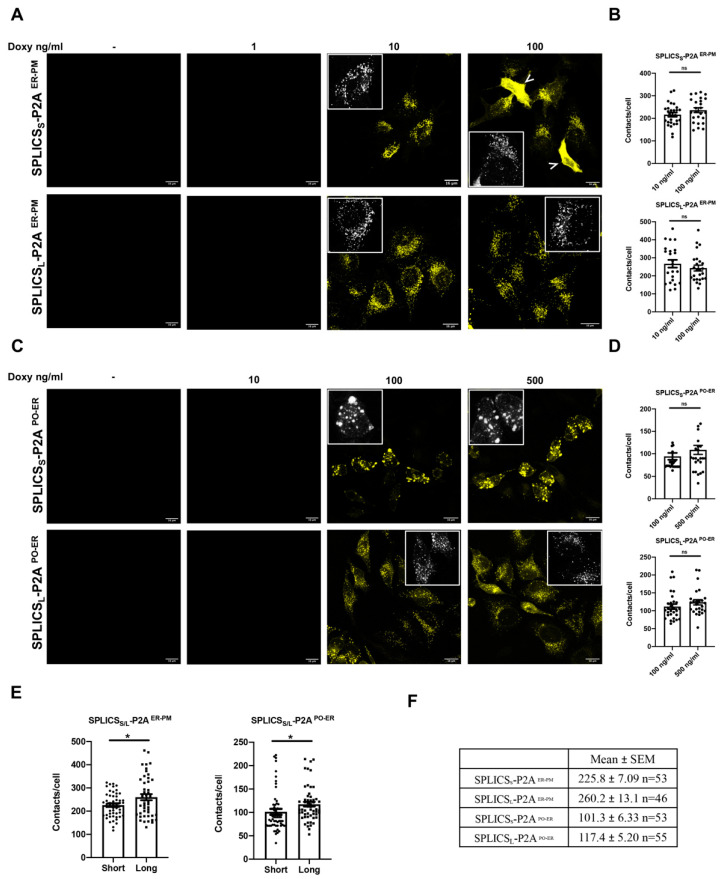
Characterization of HeLa polyclonal stable cell lines expressing SPLICS_S/L_ ^ER-PM^ and SPLICS_S/L_^PO-ER^. (**A**,**C**) Representative Z-projection images of CTRL HeLa expressing SPLICS_S/L_^ER-PM^ or SPLICS_S/L_^PO-ER^ (upon excitation at 514 nm) treated with increasing concentrations of doxycycline for 24 h. Scale bar 16 μm. (**B**,**D**) Quantification of the indicated SPLICS_S/L_ contact sites/cell was performed from the 3D rendering of a complete Z-stack at 10 and 100 ng/mL of doxycycline for ER-PM, and at 100 and 500 ng/mL for PO-ER. Data are shown as mean ± SEM dots (*n* = 3 independent experiments) and analyzed by *t*-test but no significance was detected. ns: not significant. (**E**) Quantification of SPLICS_S/L_^ER–PM^ and SPLICS_S/L_^PO–ER^ contacts by 3D rendering of complete Z-stacks, mean ± SEM of three independent experiments and analyzed by *t*-test (* *p* < 0.05). (**F**) Mean ± SEM values of the number of short and long ER–PM and PO-ER contacts. Inset: zoomed region.

**Figure 4 cells-11-01643-f004:**
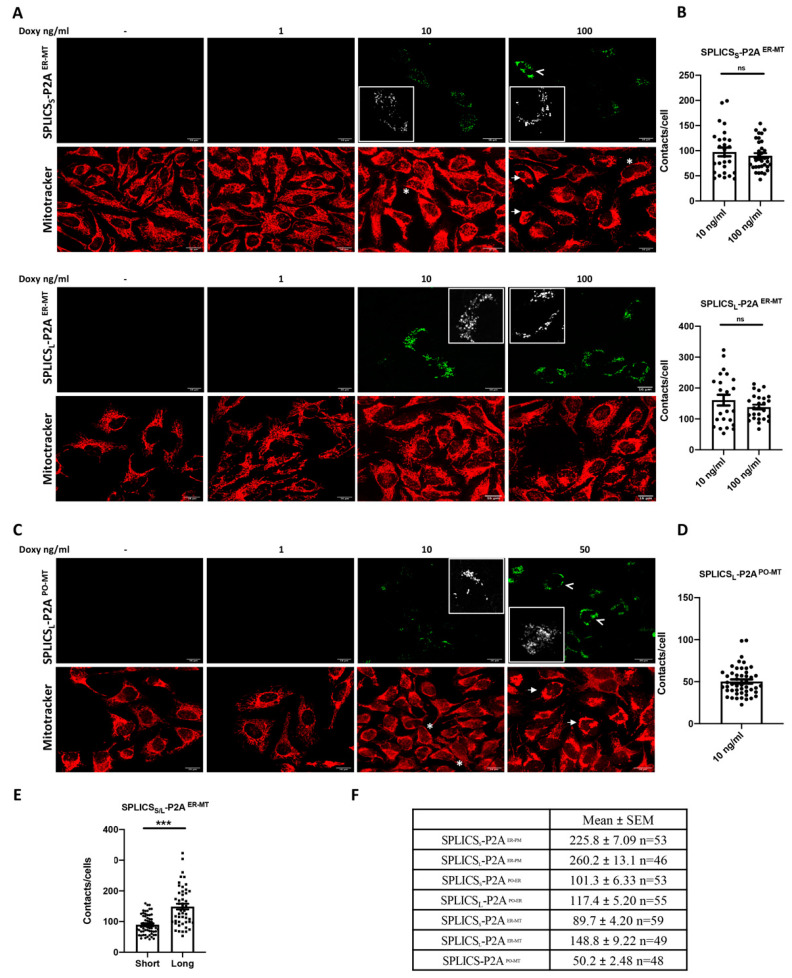
Characterization of HeLa polyclonal stable cell lines expressing SPLICS_S/L_^ER-MT^ and SPLICS_L_^PO-MT^. (**A**,**C**) Representative Z-projection images of CTRL HeLa expressing SPLICS_S/L_ ^ER-MT^ and SPLICS_L_^PO-MT^ (upon excitation at 488 nm) treated with increasing concentrations of doxycycline for 24 h. MitoTracker™ Red CMXRos (578 nm) was used to detect mitochondria morphology. SPLICS altered signal is indicated by the white arrowheads. Scale bar 16 μm. (**B**,**D**) Quantification of the indicated SPLICS contact sites/cell was performed from the 3D rendering of a complete Z-stack at 10 and 100 ng/mL of doxycycline for ER-MT_S/L_ and at 10 ng/mL for PO-MT. Data are shown as mean ± SEM dots (*n* = 3 independent experiments) and analyzed by *t*-test. Representative merge panels for SPLICS_S/L_^ER-MT^ and SPLICS_L_ ^PO-MT^ at 488/578 nm of the two higher doxycycline concentrations. ns: not significant. (**E**) Quantification of SPLICS_S/L_^ER–MT^ (both at 10 and 100 ng/mL of doxycycline) contacts by 3D rendering of complete Z-stacks, mean ± SEM of three independent experiments and analyzed by *t*-test (*** *p* < 0.001). (**F**) Mean ± SEM values of the number of ER-PM, PO-ER, ER–MT contacts, short and long, and the number of PO-MT contacts. Inset: zoomed region.

**Figure 5 cells-11-01643-f005:**
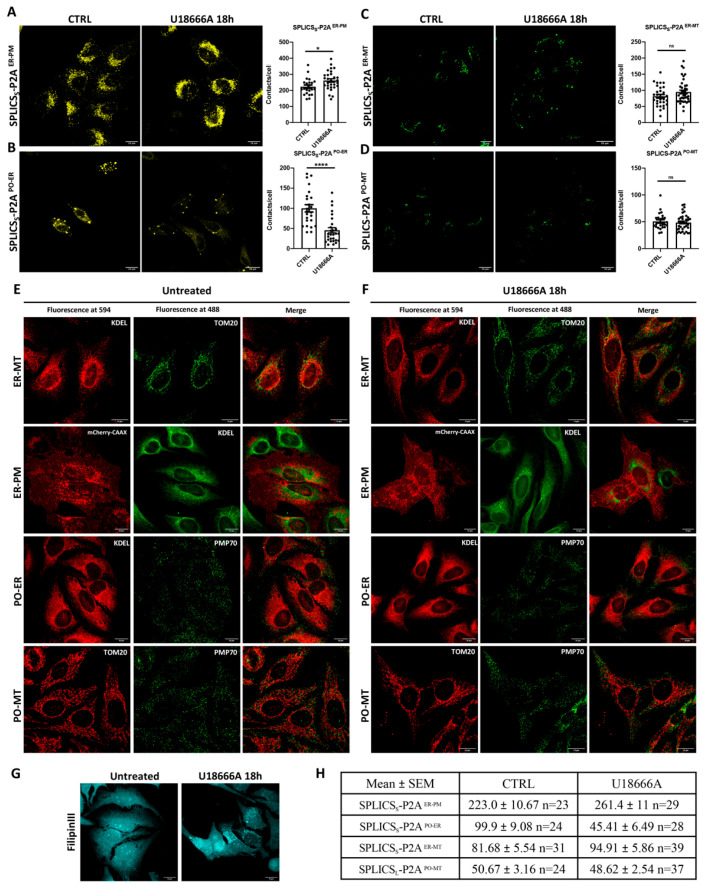
Effect of U18666A treatment in HeLa polyclonal stable cell lines expressing SPLICS_S_^ER-PM^, SPLICS_S_^PO-ER^, SPLICS_S_^ER-MT^, and SPLICS_L_^PO-MT^. (**A**–**D**) Representative Z-projection images of CTRL and U18666A-treated (2 µg/mL) HeLa expressing SPLICS probes. Cells were treated with U18666A for 18 h after 24-h stimulation with 10 ng/mL doxycycline for SPLICS_S_^ER-PM^, SPLICS_S_^ER-MT^, and SPLICS_L_
^PO-MT^, and with 500 ng/mL doxycycline for SPLICS_S_
^PO-ER^. Quantification of the indicated SPLICS contact sites/cell was performed from the 3D rendering of a complete Z-stack in CTRL and U18666A treatment conditions. ns: not significant. (**E**,**F**) Immunofluorescence analysis in cells untreated (**E**) and treated (**F**) with U18666A showing the morphology of ER and mitochondria, ER and PM, peroxisomes and ER, and peroxisomes and mitochondria. Endogenous markers used are shown. (**G**) Filipin II staining in HeLa cells showing cholesterol accumulation upon U18666A treatment. (**H**) Mean ± SEM values of the number of ER-PM, PO-ER, ER-MT, and PO-MT contacts in mock- and U18666A-treated cells. Data are shown as mean ± SEM dots (*n* = 3 independent experiments) and analyzed by *t*-test (* *p* < 0.05, **** *p* < 0.0001).

## Data Availability

The datasets generated for this study are available from the corresponding authors upon request.

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
