# Peer review of "Stable Integration of Inducible SPLICS Reporters Enables Spatio-Temporal Analysis of Multiple Organelle Contact Sites upon Modulation of Cholesterol Traffic"

_cells, 2022, doi:10.3390/cells11101643_

Round 1

Reviewer 1 Report

In the current manuscript authored by Giamogante et al., the authors have expanded on their previous work of split-GFP (SPLICS) constructs for measuring inter-organelle contact sites. They have earlier generated and validated the library of SPLICS constructs for transient transfection studies in the Vallese et al (2020) paper. In the current study, authors went further ahead and generated library of Tet-On inducible system-based SPLICS constructs for stable integration into the cells. They have validated their inducible expression, protein levels, and provide proof of concept experiments for usage in a spatio-temporal manner. The library of the SPLICS constructs generated in this study have the potential to act as valuable tools in future studying the dynamics of inter-organellar communications via membrane contact sites. Overall, I believe the study has been well designed, and executed. However, I have few minor comments which need to be addressed before the manuscript can be considered for publication. Hence, I recommend the manuscript for minor revision.

Comments:

  1. To establish the robustness of the system, the turnover/steady-state levels of the split-GFPs are to be assessed under different organelle stressors specifically in case of ER stressors, such as Thapsigargin, DTT, or Tunicamycin treatments; Mitochondrial depolarizing conditions.
  2. Its interesting to note that blocking cholesterol trafficking did not alter ERMCS, however, previous literature suggests that depleting cholesterol using Methyl-b-cyclodextrin affects ERMCS. Can authors comment on this discrepancy? Including this discussion section will be good for readers.
  3. All the microscopy images were hard to see, including but not limited to, for changes in organelle morphologies as they are too zoomed out. Can authors please provide zoomed in images with 1-3 cells in the field? Grayscale images will help visualize the punctate structures in case of peroxisomal contacts.
  4. Figure 5: Western blots are required to show no changes in GFP/YFP levels upon U18666A treatments.
  5. Figure 2: Western blots for GFPb11 fragment are required to exclude the potential changes in contacts is not due to higher levels of GFPb11 fragment.
  6. Figure 1SB&C: the blot for YFP show 2 bands. Please comment.
  7. Please rephrase the sentences line# 67-69 – Not clear.
  8. Please rephrase the sentences line# 236-238 – Not clear.

Reviewer 2 Report

Giamogante et al. describe the generation of multiple membrane contact site (MCS) sensor that are stably expressed in cell lines. These reporters are based on the highly reliable and successful SPLICS reporter system developed by the Cali lab. The new reporters cover the entire set of major MCS in the cell. In the described system, the reporters can be expressed in a dose-specific manner using doxycycline. They then use these indicators to investigate the effects of NPC1 knockdown, which led to intriguing results.

Specific points:

  1. In Figure 3A, the authors state that arrows point out cells with altered patterns at the high concentration of doxycycline. This appears to lead to artifacts that do not manifest in the number of MCS calculations, so a better way to describe the results would be that lower concentrations should be preferred.
  2. The mCHERRY CAAX signal in Figure 5E does not correspond to uniquely PM staining and should be replaced with a better indicator. Interestingly, the supplemental figure does not show this problem. Was a different indicator used for these experiments?
  3. A table summarizing the observed numbers of contacts for each individual type of MCS would help the reader understand the relative frequency of contacts. These numbers could be compared to the recent assessment from the Lippincott-Schwartz lab using a different approach.

Round 2

Reviewer 1 Report

I thank the authors for addressing my comments.  I recommend the manuscript in its current form for publication.